# A Review of Functional Outcomes after the App-Based Rehabilitation of Patients with TKA and THA

**DOI:** 10.3390/jpm12081342

**Published:** 2022-08-21

**Authors:** Henrik Constantin Bäcker, Chia H. Wu, Dominik Pförringer, Wolf Petersen, Ulrich Stöckle, Karl F. Braun

**Affiliations:** 1Charité—Universitätsmedizin Berlin, Corporate Member of Freie Universität Berlin and Center for Musculoskeletal Surgery, Humboldt-Universität zu Berlin, Charitéplatz 1, 10117 Berlin, Germany; 2Royal Melbourne Hospital, Department of Orthopedic Surgery, 300 Grattan Street, Parkville, VIC 3050, Australia; 3Department of Orthopedics & Sports Medicine, Baylor College of Medicine Medical Centre, Houston, TX 77030, USA; 4Klinik und Poliklinik für Unfallchirurgie, Klinikum Rechts der Isar der TU München, Ismaninger Street 22, 81675 München, Germany; 5Department of Orthopedics, Martin Luther Hospital, Berlin-Grunewald, Caspar-Theyß-Straße 27-31, 14193 Berlin, Germany

**Keywords:** applications, knee, hip, osteoarthritis, rehabilitation

## Abstract

Following the outbreak of SARS-CoV-2, several elective surgeries were cancelled, and rehabilitation units were closed. This has led to difficulties for patients seeking access to rehabilitation in order to achieve the best possible outcome. New applications with or without sensors were developed to address this need, but the outcome has not been examined in detail yet. The aim of this study was to perform a systematic literature review on smart phone applications for patients suffering from hip and knee osteoarthritis after arthroplasty. The MEDLINE/PubMed and Google databases were queried using the search term “[APP] AND [ORTHOPEDIC]” according to PRISMA guidelines. All prospective studies investigating rehabilitation applications reporting the functional outcome in hip and knee osteoarthritis after arthroplasty were included. The initial search yielded 420 entries, but only 9 publications met the inclusion criteria, accounting for 1067 patients. In total, 518 patients were in the intervention group, and 549 patients were in the control group. The average follow-up was 9.5 ± 8.1 months (range: 3 to 23.4 months). Overall, significantly lower A-VAS values were observed for the interventional group in the short term (*p* = 0.002). There were no other significant differences observed between the two groups. Smart phone applications provide an alternative to in-person sessions that may improve access for patients after total joint arthroplasty. Our study found there are significant improvements in the short term by using this approach. In combination with a blue-tooth-enabled sensor for isometric exercises, patients can even receive real-time feedback after total knee arthroplasty.

## 1. Introduction

Since the inception of the SARS-CoV-2 pandemic, digitalization has progressed rapidly around the world [1,2]. Social distancing has contributed to a reduction in elective surgeries in orthopedic surgery and has decreased access to postop rehabilitation for arthroplasty patients [3]. This may have led to worse outcomes and delayed returns to activity. Additionally, restrictions in the range of motion are likely to be present, resulting in major limitations. Although digital surgical approaches involving virtual and augmented reality, as well as robot navigation systems, have become increasingly popular, there is a lack of digital rehabilitation programs. With the number of smartphone users worldwide estimated to reach 3.8 billion by 2021 [4], there is an opportunity to deliver physical therapy easily at relatively low costs. Bahadori et al. recently published a systematic review, showing that there are 15 applications available on smartphone app stores focusing on rehabilitation following total hip and knee arthroplasty. However, these varied significantly in their quality and outlined a missing partnership with patients. Therefore, the authors questioned the clinical importance [5]. In a further study, the evidence supporting the use of smartphone apps and wearable devices was assessed. Here, wearable devices were capable of monitoring physical activity and improving patient engagement following total knee arthroplasty [6]. Both of these reviews, however, failed to report the functional outcome in detail.

In addition, telemedicine, virtual digital scribes, chat bots and surgical scheduling applications for consultations have all reportedly led to new improvements in patient care [7]. For digital rehabilitation, telehealth visits with a live video feed in conjunction with a physical therapist are still the most common. To date, physical therapy delivered entirely via smart phone applications lacks evidence. In 2019, Campbell et al. showed that patients receiving automated text messages after total joint arthroplasty led to an increased amount of time spent on home exercises. This also leads to improvements in the patient’s mood and a decreased use of narcotics, while minimizing calls to the surgeon’s office [8].

Besides home training reminders delivered via text messages, an app-based approach offers a variety of different individual training programs; this can allow a blue-tooth-enabled sensor to be connected to provide real-time feedback. This has the potential to improve compliance, as it is more convenient logistically, especially for immobilized patients. Furthermore, cost and wait times can be reduced. This study aims to summarize the published data on using this approach to provide postoperative rehabilitation for hip and knee arthroplasty patients.

## 2. Materials and Methods

A systematic literature review was conducted on 25 April 2022, searching the MEDLINE, Cochrane and Google scholar databases. The Preferred Reporting Items for Systematic Reviews and Meta-Analyses (PRIMSA) guidelines were followed (Figure 1) [9].

The broadest inclusive terms were applied to include all relevant studies. We used the search term “[APP] AND [Orthopedic]” and included all full articles in English, German and French. The analysis was performed by a fellowship-trained orthopedic surgeon and a sports medicine physician. We only included prospective studies that investigated application-based rehabilitation programs in patients suffering from knee and hip osteoarthritis, as well as those who had undergone total hip and knee arthroplasty. We excluded retrospective studies, letters to the editor and comments or studies that are not full articles. Studies that lack a functional outcome were also excluded. 

For each study that met the inclusion criteria, the number of patients, gender, age, comorbidities, length of hospitalization, follow-up, body mass index (BMI), indication (knee vs hip osteoarthritis), patient-reported outcome measures (PROM) and apps used with or without a sensor were analyzed. Regarding functional outcome, changes in endurance tests within six minutes, changes in walking speed, changes in functional mobility, the time up to go (TUG) test, the five times sit to stand test (FSST), the hip disability and osteoarthritis outcome score (HOOS), the knee injury and osteoarthritis outcome score (KOOS) and the SF-36 survey were recorded. Other outcome measures include the Knee society score (KSS), visual analogue scale at rest as well as activity (R-VAS/A-VAS), range of motion (ROM), Western Ontario, McMaster Universities Osteoarthritis Index (WOMAC), Euro Quality of Life (EQ-5D-3L), short questionnaire to assess health-enhancing physical activity (SQUASH) and patient activation measure (PAM-13).

For statistical analysis, SPSS (SPSS, Inc., IBM Company, Chicago, IL, USA) and Microsoft Excel (Microsoft Corporation, Redmond, DC, USA) were used. All data are presented in absolute numbers and percentages; significances are set to *p*-values < 0.05. Because of the heterogeneity of the data, we were not able to perform a meta-analysis. 

## 3. Results

Our search terms revealed a total of 420 entries. A total of 99 publications investigated the application-based rehabilitation, of which 36 publications investigated either hip or knee osteoarthritis in the context of total knee or hip arthroplasty. A total of 27 articles were excluded, leaving 9 publications for the final analysis.

Within the 9 studies, 1067 patients were analyzed, including 518 patients in the intervention group and 549 patients in the control group. The mean age was 63.3 ± 3.5 years, and 41.8% of patients were male (*n* = 384/919). The mean BMI was 29.5 ± 2.3 kg/m^2^, and patients suffered from 3 ± 2.3 comorbidities, on average. The mean length of hospital stay was reported to be 9.1 ± 4.2 days, and the average follow-up was 9.5 ± 8.1 months (range 3 to 23.4 months). For rehabilitation programs, four authors reported that a sensor or motion tracker was used, whereas an app-based exercise instruction was delivered via a smartphone or a tablet in the remaining five studies. All of the included studies are illustrated in Table 1.

For total knee arthroplasty, four studies were included [10,11,12,13]. Three of the four studies reported the KOOS score [10,11,12]. Two of the studies reported the long-term outcome, whereas only one reported the functional outcome after a one-week follow-up [11]. Van Dijk-Huisman reported the standing and walking times and the functional recovery [13]. Only one study presented the TUG, HOOS and SF-36 scores [14]. In the remaining four studies, patients suffered from knee [15] and hip osteoarthritis [16,17,18]. Those studies reported the changes in the functional mobility, timed up to go test, disability and functional independence measure [16], self-management behavior [17] and visual analogue scale at rest (R-VAS) [18]. 

Although higher KOOS scores were found in the short term as well as the long term, no overall significances were found. Following THA, Wijnen described significantly higher HOOS scores for function in sport and recreational activities and hip-related quality of life. This is also observed for the SF-36 physical role limitations at 12 weeks and 6 months post-surgery [12]. The overall A-VAS score was significantly lower in the app-based group in the short term (*p* = 0.002). Notably, both of these studies were published by the same group of authors [10,11]. All PROMs and results are presented in Table 2. Furthermore, no significant differences between the individual indications and the application with or without sensors were identified.

## 4. Discussion

This review shows that patients can benefit from digital apps for the rehabilitation of osteoarthritis in the context of total joint arthroplasty, especially for the short term (A-VAS). In the long term, no statistical significances were observed, although the values for the app-based groups were slightly higher overall. Despite the different applications investigated, with or without the use of sensors, no correlations were found.

For rehabilitation, total knee arthroplasty is easier to investigate, as weakness in the flexors or extensors can be measured. The hip is much more complex, as the rotators, flexors, extensors and abductors all need to be strengthened and can be difficult to assess individually. However, overall function can still be assessed, with the six-minute endurance test and walking speed as examples. 

The development of applications and digital platforms is typically associated with high fixed costs and lower variable costs. Once an application is developed and operational in a market, the marginal cost of treating an additional patient is quite low and thus more scalable. However, insurance coverage for app-based therapy is still very limited [19]. 

Although significant improvements were observed in the short term, no significant differences were found in the long term. Digital apps can be another tool when there is no access to in-person rehabilitation following total knee arthroplasty. The addition of blue tooth sensors in total knee arthroplasty can provide real-time feedback to patients in order to motivate them. Other indications have already been investigated in sports medicine and following cruciate ligament replacements [20,21]. In particular, isometric exercises can be effectively performed this way. Eccentric and concentric movements are much more difficult to simulate, as they typically require in-person supervision. Sensors can also detect dynamic valgus malalignment, which is one of the main risk factors for anterior cruciate ligament injury and re-injury [22].

A disadvantage of this approach is that the history of injuries, previous surgeries or comorbidities may not be built into the workout program. Likewise, other factors such as surgical techniques (e.g., cemented vs. non-cemented THA, CR TKA vs. PS TKA) are typically not taken into account. Furthermore, the poor fitting and calibration of sensors can potentially provide misleading feedback for patients. This may call for a combined approach where initial sessions are instructed in person and then transition to an app-based approach with careful remote monitoring by a qualified provider to adapt the training as required. While regular text messages or push notifications can help to improve compliance, some degree of motivation is still required to use the app effectively [8]. Finally, it must be mentioned that elderly patients may have difficulty navigating smart devices and may not be willing to use applications if they prefer in-person sessions.

The app-based delivery of care is promising, and the data summarized in our study support its use. With the help of artificial intelligence and a thorough initial assessment regarding individual comorbidities, individual limitations and preoperative conditions, a personalized training/rehabilitation algorithm can be developed. This will improve the effectivity of training and, subsequently, compliance and satisfaction. In conjunction with robotic navigation, augmented reality, virtual reality and artificial intelligence can potentially optimize outcomes and improve efficiency. The different questionnaires and patient-reported outcome measures seen in the various studies included in our systemic review are difficult to compare. In addition, an individual’s motivation and compliance to app instructions are not measured in these studies. As such, it is difficult to delineate exactly how often the individual apps were used by the patients.

There are several limitations to this study. As this one is a review, the analysis depends on the individual studies. Additionally, as we only searched the PubMed/Medbase and Google Scholar databases, some articles that were published on EMBASE or the Web of Science, along with grey literature, may have been missed. Further, no quality assessment was performed. For outcomes, the rehabilitation monitored was not further specified, and, subsequently, no recommendations were made. Finally, as the data were heterogenic, we were not able to perform a meta-analysis.

## 5. Conclusions

Digital applications provide a good adjunctive tool for rehabilitation following total joint arthroplasty, with significant improvements in the short term. After total knee arthroplasty, isometric exercises in particular can be performed with a sensor to allow for real-time feedback. Eccentric and concentric exercises are more difficult to perform via this approach. We do not believe that this new technology replaces physiotherapists, but it will more likely serve as another tool in our armamentarium. With proper instruction, applications can help and motivate patients, ideally with regular follow-ups to see if adjustments are required.

## Figures and Tables

**Figure 1 jpm-12-01342-f001:**
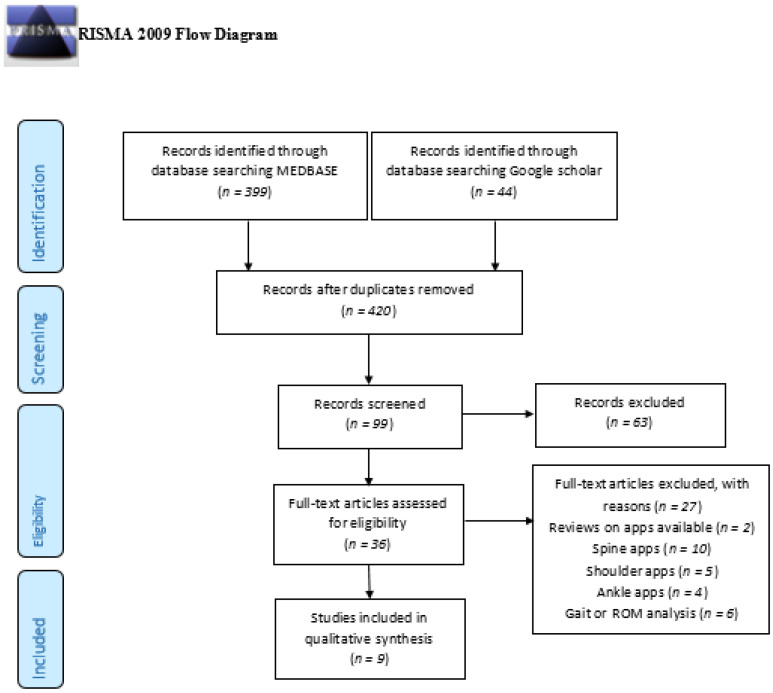
Included articles according to the PRISMA Guidelines.

**Table 1 jpm-12-01342-t001:** Included studies and demographics of patients included.

Author	Year	No. Patients	Age (Year)	Gender (Male) (No.)	BMI (kg/m^2^)	Follow-Up (Months)	Digital Program	Indication	Hospital Stay (Days)	Days of Intervention	Comorbidities (No.)
Arfaei Chitkar SS	2021	31	57.84		27.97		Application instruction	Knee osteoarthritis		56	7
Control group		29	58.52		26.62					56	5
Bäcker HC	2021	20	62.95	8	32.33	23.73	Application with sensor	TKA		6.6	1
Control group		15	66.27	6	33.79	23.35				6.9	1
Correia FD	2019	38	67.3	32	31.0	6	Application with motion tracker	TKA	6.0	56	Detailed listing
Control group		31	70.0	22	30.8	6			6.0	56	Detailed listing
Hardt S	2018	22	63.3	10	31.6	7	Application with sensor	TKA	6.6	6.6	1
Control group		25	67.6	10	32.4	7			6.9	6.9	1
Li I	2020	44	65				Application instruction	Arthritis	14.7	14.7	4
Control group		44	66						14.2	14.2	4
Pelle T	2020	214	62.1	67	27.8		Application instruction	Osteoarthritis			
Control group		213	62.1	54	27.3						
Skrepnik N	2017	107	61.6	48	29.4	3	Application with sensor	Osteoarthritis			
Control group		104	63.6	57	29.3	3					
Control group		61	66.0	40	27.73						
Van Dijk-Huisman HC	2020	27	65.1	15	27.47		Application with sensor	TKA		31	
Control group		61	66.0	40	27.73					31	
Wijnen A	2020	15	59.3	5	26.7	6	Application instruction	THA		84	
Control group		15	59.3	5	28					84	
Control group		12	59.3	5	31.1					84	
Intervention group		518	62.7 + 2.9	443 (41.8%)	29.3 + 2.1	9.2 + 8.3			9.1 + 4.9		3.3 + 2.9
Control group		549	63.9 + 3.9	467 (41.8%)	29.7 + 2.5	9.8 + 9.2			9.0 + 4.5		2.8 + 2.1
*p*-value			0.243	0.449	0.368	0.454			0.493		0.393

**Table 2 jpm-12-01342-t002:** PROMs following rehabilitation programs for knee/hip osteoarthritis, as well as following joint replacement. Score indicates the PROM used; overall means, standard deviations and *p*-values were calculated for HOOS/KOOS and self-management behavior scores, as these are comparable. Empty cells—not reported in the individual studies. N/A—not applicable. Significances presented in bold.

	No. Patients	Six-Minute Endurance	ST	Walking Speed (10 min)	ST	Change in Disability	ST	Score	Pain	ST	LT	Symptoms	ST	LT	Function in ADL	ST	LT	Sport/Recreation	ST	LT	Quality of Life	ST	LT	R-VAS	ST	LT	A-VAS	ST	LT
Arfaei Chitkar SS	31							WOMAC	18.5			5.6			48.3														
Control group	29							WOMAC	17.2			5.6			49.6														
Bäcker HC	20			11.77	19.66			KOOS	42.4	56.0	81.7	50.6	61.7	64.5	45.8	54.6	77.2	16.0	9.5	48.6	18.4	26.7	68.4	3.6	2.7	0.9	7.6	4.0	2.7
Control group	15			12.39	27.08			KOOS	37.8	53.2	80.7	48.6	63.1	61.6	40.8	51.0	77.1	9.3	8.1	47.3	14.2	30.8	67.9	4.3	3.6	0.9	6.7	5.1	2.8
Correia FD	38							KOOS	33.0	95.5	100.0	34.0	51.5	96.0	34.0	93.0	97.0	0	30.0	42.5	13.0	81.0	94.0						
Control group	31							KOOS	47.0	86.0	86.0	50.0	82.0	86.0	41.0	87.0	87.0	5.0	20.0	20.0	25.0	56.0	63.0						
Hardt S	22			0.9	0.6			KOOS	45.0	56.0		62.0	60.0		47.0	54.0		14.0	10.0		17.0	30.0		4.0	2.0		7.0	4.0	
Control group	25			0.8	0.5			KOOS	36.0	50.0		58.0	59.0		39.0	42.0		12.0	5.0		18.0	26.0		4.0	4.0		7.0	5.0	
Li I	44	delta = 115.6		delta = 0.4		17.3	118.2																						
Control group	44	delta = 103.3		delta = 0.4		18.0	117.5																						
Pelle T	214							Self-management behavior	57.5	59.5	59.4	57.7	57.3	57.3	58.5	61.4	62.1	32.6	31.9	33.4	38.0								
Control group	213							Self-management behavior	58.2	57.4	57.5	57.0	56.2	55.2	59.4	58.5	58.6	32.5	33.2	33.2	38.3								
Skrepnik N	107	402.8	18.2																					4.6		delta = −55.3			
Control group	104	395.6	6.3																					5.1		delta = −33.8			
Van Dijk-Huisman HC	27							Standing walking time/functional recovery							70.9			−0.3											
Control group	61														103.0			1.0											
Wijnen A	15							HOOS	48.9	88.8	98.7	50.0	75.3	91.0	52.7	76.5	96.8	23.3	70.0	82.5	19.2	50.8	88.8						
Control group	15							HOOS	35.5	71.7	85.1	29.3	68	76.3	34.0	60.6	80	16.3	26.7	59.6	22.9	45.8	71.3						
Control group	12							HOOS	36.3	73.5	85.6	41.7	62.1	77.5	37.1	58.1	79.1	20.8	29.7	64.9	24.5	43.2	69.3						
Intervention group	518	402.8	18.2	6.34	10.13	17.3	118.2		45.36	71.16	84.95	50.86	61.16	77.2	47.6	67.9	83.275	17.18	30.28	51.75	21.12			4.07	2.35	0.9	7.3	4	2.7
Std.		N/A	N/A	7.69	13.48	N/A	N/A		8.97	19.36	18.96	10.68	8.80	19.16	9.13	16.71	16.90	12.06	24.61	21.43	9.73			0.50	0.49	N/A	0.42	0	N/A
Control group	549	395.6	6.3	6.60	13.79	18.0			41.80	65.30	78.98	47.43	65.07	71.32	41.88	59.53	76.36	15.98	20.45	45	23.82	40.36	67.88	4.47	3.8	0.9	6.85	5.05	2.8
Std.		N/A	N/A	8.20	18.79	N/A	N/A		9.12	14.00	12.20	10.70	9.20	12.58	8.97	15.09	10.61	9.77	11.65	18.57	8.23	12.04	3.54	0.57	0.28	N/A	0.21	0.07	N/A
*p*-value		N/A	N/A	0.977	0.844	N/A	N/A		0.532	0.574	0.583	0.609	0.493	0.595	0.324	0.405	0.475	0.859	0.405	0.628	0.630	0.607	0.069	0.413	0.069	N/A	0.312	**0.002**	N/A

## Data Availability

Not applicable.

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
