# Peer review of "A Review of Functional Outcomes after the App-Based Rehabilitation of Patients with TKA and THA"

_jpm, 2022, doi:10.3390/jpm12081342_

Round 1

Reviewer 1 Report

The aim of the study is to investigate the results of application-based rehabilitation in patients with osteoarthritis of the hip and knee after arthroplasty. This topic is very interesting and important nowadays. However, I think some modifications are needed.

Introduction

There is a lack of information on why physical therapy is so crucial for hip and knee arthroplasty patients.

Materials and Methods

A major concern is that the method for selecting the articles included in the review is not clear. First, the flow chart presented in Figure 1 should be improved. There is no information on the number of records obtained from each database. Second, the chart also does not provide information on the reasons for excluding articles. Did you evaluate the quality of the included articles?

I think that the characteristics of the statistical analysis should be more detailed, especially in terms of meta-analysis.

Results

Another major concern is that tables 1 and 2 are quite confusing. Please improve the readability of the tables. The order of presentation seems random (i.e., neither alphabetical nor chronological). Maybe it would be better to present the author with the date in a separate column and group in the next column. Please be consistent and systematic in providing other characteristics. What do empty cells mean in tables? This needs to be clarified. I assume this means something has not been investigated or reported in the study.

The abbreviations in the tables are not explained, and no units were given. What does it mean e.g. “symptoms/other”? Be more consistent in reporting decimal places.

Since differences in patient-reported outcome measures make statistical comparisons difficult, you can focus more on narrative review. In my opinion,  it would be interesting to report what kind of rehabilitation was provided by apps, what were the duration and frequency of therapy or it was just monitoring of physical activity? How physical activity was measured in detail? What feedback did patients and therapists receive? Was app rehabilitation monitored by a physiotherapist at all?

Discussion and Conclusions

Some discussion of the beneficial improvements in rehabilitation apps might be worthwhile. What do you think, how app-based orthopedic rehabilitation can be more personalized?

Author Response

Thank you so much. We really appreciate your help to improve our manuscript. We made all effort to comply with your comments.

Reviewer 2 Report

Dear Authors,

thank you for the chance to review the manuscript.
Overall, the review is of relevant quality.  According to PRISMA this is not a Systematic Review and for sure no Meta-Analysis.
Please find details for review below:

Title: Please rethink, for example: "A Review of functional outcomes after App based rehabilitation of patients with TKA and THA" or something similar would rather meet the content.

Abstract:
According to PRISMA 2020 for Abstracts for example Risk of Bias tool and Preregistration are mandatory for SRs. Please double check

Introduction: Rational and objective are explained. Still, there are a few reviews on similar topics out there. Please consider citing and mentioning what they are missing to report - leading to the rationale for this review.

Methods
PubMed is not a database. Medline can be assessed via PubMed or OVID. CL is a library for SRs. This leads to the problem that only Medline as a database was reviewed for primary medical articles. I guess Google scholar is supposed to be the google database. If yes, please rewrite. So this methodological flaw might be solved. 
Especially in combination with the search string this remains discussable. This review probably missed quiet some articles. An extended search string or a MESH Search on the terms [APP] OR [mobile] AND [[Orthop*] OR [TKA]..... and in more databases (e.g. EMBASE, Web of Science and grey literature) would lead to more sufficient results. Please state this as limitation.

If the study is preregistered on PROSPERO, please reference preregistration link for transparency. The flowchart is fine.

It seems there was no Risk of Bias tool used, what is a major problem in SRs!?

The rationale for an Meta-analysis  should be explained: clincal and statistical Heterogeneity, Sensitivity analysis, but it seems like, no meta-analysis was performed, at least I can not find any forest plot? 

Even if this literature overview is a really relevant topic, it should not be published as a Systematic Review. There are to many essential points missing, like the databases, sufficient search string, critical appraisal, Forest Plot.... 

Please consider to publish as a narrative review.

If published as a narrative review, results and discussion are fine. If it should be published as a SR with MA intensive revision would be needed.

Author Response

Many thanks for your help and comments. We revised our manuscript accordingly and changed the title as recommended.
